# Design and Evaluation of a Personal Robot Playing a Self-Management for Children with Obesity

**Tareq Alhmiedat** [1,2,*] and **Mohammed Alotaibi** [3]

1   Department of Information Technology, Faculty of Computers & Information Technology, University of Tabuk, Tabuk 71491, Saudi Arabia

2   Industrial Innovation & Robotics Center, University of Tabuk, Tabuk 71491, Saudi Arabia

3   Department of Computer Science, Faculty of Computers & Information Technology, University of Tabuk, Tabuk 71491, Saudi Arabia

*   Correspondence: t.alhmiedat@ut.edu.sa

**Abstract:** The preponderance of obesity and being overweight among children has increased significantly during the last two decades in Saudi Arabia and United Arab Emirates (UAE) with overwhelming consequences to public health. Most recommended approaches have paid attention to a healthier diet and physical activity (PA) to reduce obesity. Recent research shows that the use of social robots could play a vital role in encouraging children to improve their skills in self-management. As children need to be surprised and feel a sense of enjoyment when involved in any activity where they can spend time and actively engage in activities, social robots could be an effective intervention for this purpose. In this context, the current project aimed to build an innovation social robot system to offer a set of activities to help obese children improve their capabilities to manage their selves properly and increase their obesity knowledge. This study aimed to determine the perceptions of obese children towards the NAO robot, a new medical technology, and analyze their responses to the robot's advice and education-related activities. A proposed model of the intervention using the NAO robot is discussed in this study, and a pilot study was conducted to assess the performance of the proposed system. The obtained results showed an average acceptability of 89.37% for social robots to be involved in obesity management.

**Keywords:** obesity self-management; robot assistance; social robots; childhood obesity

## 1. Introduction

During the last two decades, due to changes in people's lifestyle, obesity has become one of the most common chronic diseases globally. Obesity is measured by using the body mass index (BMI) value. BMI is defined as "the body mass divided by the square of the body height, and is universally expressed in units of $kg/m^2$, resulting from mass in kilograms and height in meters" [1]. According to recent statistics, worldwide, 38.9% of the adult population, 17.3% of adolescents (10–19 years), 20.6% of school-age children (5-years), and 5.9% of preschool children (less than five years) were identified to be overweight. Furthermore, the share of the global overweight population is projected to reach 42% by 2025. These statistics present a serious global health concern (overweight), as it could lead to various health risks, which can significantly affect public health globally and can create a huge burden on already stressed-out healthcare systems across the globe [2]. Obesity, which was once considered to be more prevalent in high-income countries, has been slowly increasing its prevalence in low-middle income countries.

Almost half of the global obese children under the age of 5 were found in Asia alone, and the prevalence of diabetes among children and adolescents has risen from 4% in 1975 to 18% in 2016 [3]. Saudi Arabia is one of the countries in the Middle East where the prevalence of obesity is very high (45%) among adults and high (25.3% and 23.26%) among children aged between 5–9 years and 10–19 years, respectively. It is interesting to note that

the annual increase in the obesity rate among children is high (2.7%) compared to that of adults (1.6%), reflecting a serious concern for children's health in Saudi Arabia [4].

According to a recent study [5], the kingdom of Saudi Arabia is now among the nations with the largest obesity and overweight prevalence rates because of a number of factors. The main factors involve eating habits, marital status, genetic factors, diet pattern, family history, and the lack of physical activities, whereas the major consequences include diabetes, ischemic heart disease, and cancers. Obesity is a major concern in the kingdom of Saudi Arabia, as 7 out of 10 people are considered obese or overweight [6]. The economic impact of obesity and overweight in Saudi Arabia in 2019 was identified to be USD 19 billion (2.42% of GDP), and it is projected to reach USD 78 billion (4.16% of GDP) by 2060, reflecting a significant rise in healthcare costs [4]. Many studies [7–9] reported that there is a relation between obesity and other chronic diseases, such as diabetes and hypertension. Therefore, there is a high risk of getting diabetes among children in their early ages. Modern lifestyle changes, less physical activity, and dietary habits are considered the most important reasons behind the high rates of obesity and being overweight among children in Saudi Arabia [10]. Therefore, it is very important that efforts are put in place by healthcare decision-makers in Saudi Arabia to create awareness among children about the risks of obesity and to promote self-management practices in managing obesity and maintaining healthy lifestyles.

Consequently, the use of modern technologies such as Internet of Things, mobile health, and social robotics could be an excellent and cost-effective solution to improve society's awareness and healthcare education and to improve the innovative technology infrastructure in the Saudi Arabian healthcare system, supporting the objectives of Vision 2030 in transforming the traditional healthcare system into a digitized healthcare system. Furthermore, it can address the issue of a lack of special centers for obesity and diets, especially in the remote areas of Saudi Arabia. These are a few factors that motivated the authors to develop an innovative robotic platform to enhance the self-management practices and awareness of obesity among children. Moreover, in Saudi Arabia and other Gulf countries, there has not been any social robot system identified to enhance obesity management and awareness among children. Therefore, unconventional strategies such as robotic interventions are required on an urgent basis to deliver health services and medical education to address the needs of obese and overweight children in an innovative, interesting, and cost-effective way.

The rapid development of telecommunication and Internet of Things technologies, smart mobile devices, and robotic technology has accelerated the design and implementation of healthcare service delivery for the management and awareness of various chronic diseases, such as obesity and diabetes [11–15]. In developing any technology intervention for children, it is important to understand that children's requirements are different from adults, as they like things that are attractive, that grab their attention, and that surprise them. The use of robotic technology could be an effective intervention, as it can fulfill children's needs by using social robots.

Chronic disease management, such as diabetes, can be improved by the use of social robotics. A systematic review study [16] focused on the role of using social robotics in improving diabetes management among children, and it identified six studies that focused on the use of social robotics in improving diabetes management, diabetes awareness, or both diabetes management and awareness in different counties. The study concluded that the use of robotic technology played a significant role in improving diabetes awareness and management among children.

Several recent studies discussed the greater acceptance of social robots by children. According to a recent study [17], authors measured the acceptance of an international robot, using the measurements enjoyment, social presence, and social anxiety among 87 children aged between 7 and 11 years.

On the other hand, authors of [18] investigated the acceptance of social robots by conducting three interventions (54 children randomly selected for one of the three interventions): interactive with social robot teddy bear, tablet-based avatar version of the

bear, and plush teddy bear with human presence. The authors revealed that social robots appeared to be an engaging tool that may offer new ways to address the emotional needs of hospitalized children. Similarly, another study [19] analyzed potential experiences of children's intentional and behavioral aspects, revealing social robot acceptance.

Considering these factors, the aim of this study was to determine the perceptions of obese children towards the NAO robot, a new medical technology, and analyze their responses to the robot's advice and education-related activities, and the following objectives were outlined for achieving the aim of this study:

1. Develop a social robot architecture to deal with children with obesity.
2. Validate the acceptability of social robots with children with obesity through a pilot study.
3. Discuss the results obtained from the pilot study.

The remainder of this paper is organized as follows: Section 2 discusses the recent developed obesity management systems targeted the children. The proposed obesity management system is presented in Section 3, where Section 4 discusses the experiment testbed and the obtained results. And finally, Section 5 concludes the work presented in this paper and presents future works.

## 2. Related Works

In the past two decades, tremendous upgrades in portable and sensor innovation have permitted numerous investigations on the use of versatile wellbeing, telehealth, and robotic interventions in medical care. Portable wellbeing/telehealth and electronic frameworks give a programmed or self-loader uphold instrument for ongoing illness among patients, empowering far-off patient expert consideration and backing remedial outcomes by medical care experts to their patients. However, a large number of studies did not focus on utilizing versatile wellbeing innovation and robot gadgets to improve persistent infections, for example, hypertension and diabetes. However, the use of social robots in improving obesity among kids in the literature is very limited.

A recent study [20] presented a platform for childhood obesity prevention based on interactive games played between a social robot and the child. They depended on the use of the Anki© Cozmo robot. The authors focused on four distinct stages: (a) familiarization with the robot, (b) education and knowledge assessment (e.g., for a balanced diet), (c) behavioral data collection and empowerment, and (d) personalized goal-setting. In the familiarization stage, face-recognition technology was used for the robot to recognize the face of the child (face/object recognition), and it proceeded with a sound, kinetic, or facial reaction. In the education stage, a series of interactive health educational games related to obesity and diet were played by the robot and the child. In the behavioral data collection and empowerment stage, two questionnaires, Food Frequency Questionnaire (FFQ) [20] and the Physical Activity Questionnaire for Children (PAQC) [21], were used to collect the data.

In the final stage of personalized goal-setting, after processing the input behavioral data received within stage 3, the child was advised by the robot (through speech) to follow one specific goal (goal selection engine) for diet or physical activity for the following week. The platform was evaluated in terms of usability, with 10 participants achieving a system usability score (SUS) of 93.8. This study concluded the high potential of social robots for promoting healthy behaviors and preventing childhood obesity. The study was later updated and published in 2020 in a different version [22] by adding a fruit recognition model, which was based on a deep learning model based on Google Inception V3, applied for the use case of image-based fruit recognition. Using this model, an accuracy of 99.68% was achieved.

To the best of the authors' knowledge, this is the first study of its kind that aimed to employ social robots with children with obesity in order to improve the self-management behavior of the children.

### 3. Obesity Management System Design

The advanced mechanic's field had a binding effect in each medication zone, including assisting medical procedures, apportioning medicine, sanitizing rooms, etc. This section discusses the social mechanical frameworks for managing obesity among children. As presented in Figure 1, the proposed obesity management framework comprised three principal areas, which included child side, webserver side, and medical staff-side.

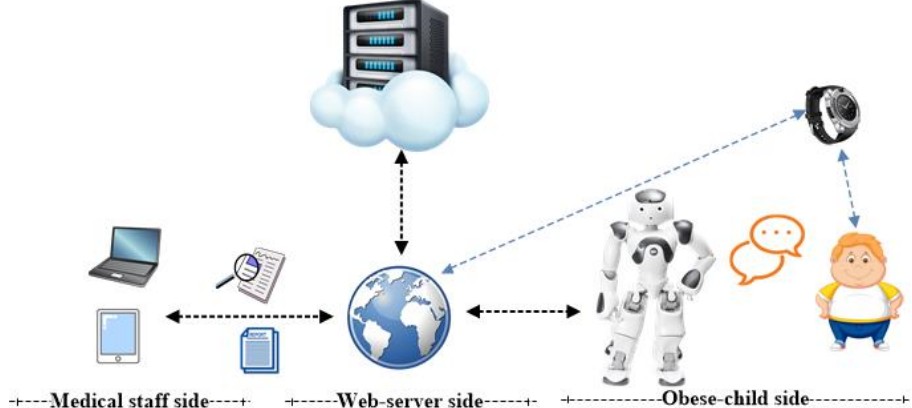

+----Medical staff side----+ +------Web-server side------+ +-----------Obese-child side-----------+

**Figure 1.** The obese management system concept.

#### 3.1. System Design

The obese-child side consisted of the robot, obese children, and the smart monitoring unit, as presented in Figure 1. The smart monitoring unit collected activity information from the obese children and transmitted it to the webserver. On the other hand, the webserver side involved the children's activities and general statistics about the child interaction with the robot. On the medical-staff side, a medical person received the child's activities from the smart monitoring unit and also received general statistics about the child's communication with the robot.

The developed robotic obesity management system consisted of four main modules, as presented in Figure 2, which include monitoring the child's activity, collecting activity data, analyzing the collected data, and uploading the data on a webserver.

- Monitoring child-activity: This includes a smart-watch unit that incorporated an intelligent system that could monitor, record, and transmit the child's activities during the daytime. The child's general activities (active/inactive, number of steps, and the activity in the whole daytime) were collected and processed on the monitoring unit.
- Upload activity data on a webserver: The processed data were then transmitted to the firebase webserver to allow this data to be available for the medical staff and the robot system.
- Analyzing the collected data: The collected data were processed and analyzed according to the previous child's activities recorded in the database and according to the child's general status.
- Robot collection of child-activity: In this stage, the robot collected the analyzed data and interacted with the obese child.

The developed obesity management system monitored the child's activities during the daytime using the smart-watch device. The smart-watch device was equipped with wireless communication to transmit the collected activity data from onboard sensors to the firebase webserver. Figure 3 shows the flowchart for the monitoring unit, where the collected data from the smart watch were recorded and transmitted to the webserver, where the webserver made this data available for the NAO robot and medical staff.

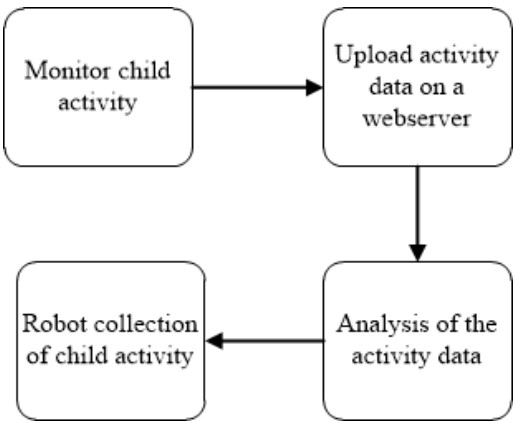

**Figure 2.** Obesity management functions.

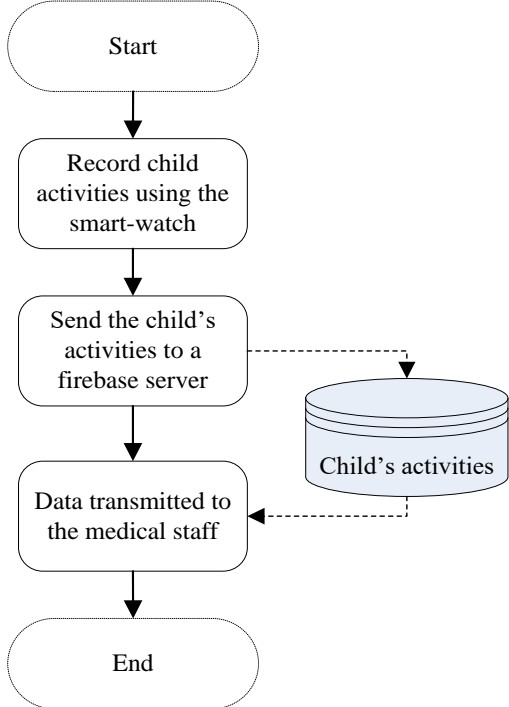

**Figure 3.** Flowchart for the smart-watch device function.

On the other hand, the robot interacted with the obese child by discussing the current state of the child's progress and advised the child regarding his/her recent activities. The proposed obesity management system managed the obesity in children in two main ways: first, the monitoring unit, and second, the education through the quiz and stories section.

Through the interaction process, the robot first identified the child. Then, based on the child's identity, the robot collected the child's activities for the last few days, and it compared the obtained activity results with the previous results. The robot then advised the child after receiving and processing the activity data from the webserver. It is important to note that the robot behaved according to the status of the child's activities, enabling an emotional connect with the children. Figure 4 shows the flowchart for the interaction process between the obese child and the NAO robot.

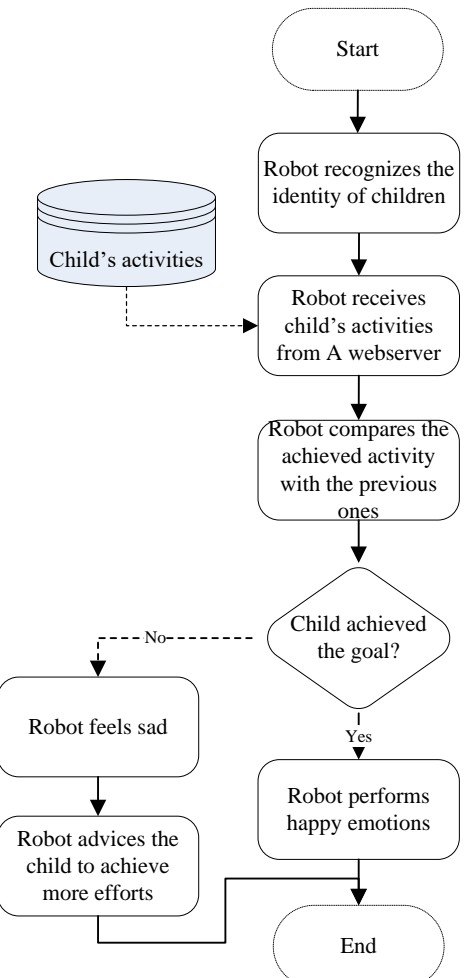

**Figure 4.** Flowchart for the interaction process between the obese child and NAO robot.

The interaction process between the obese children, medical staff, and the robot is illustrated in Figure 5. The smart-watch device collected the activity data from the obese children and then transmitted them to the webserver. The medical staff was able to retrieve reports regarding the child's activities and take a suitable decision. On the other hand, the robot received the child's activities as soon as the robot interacted with the obese child.

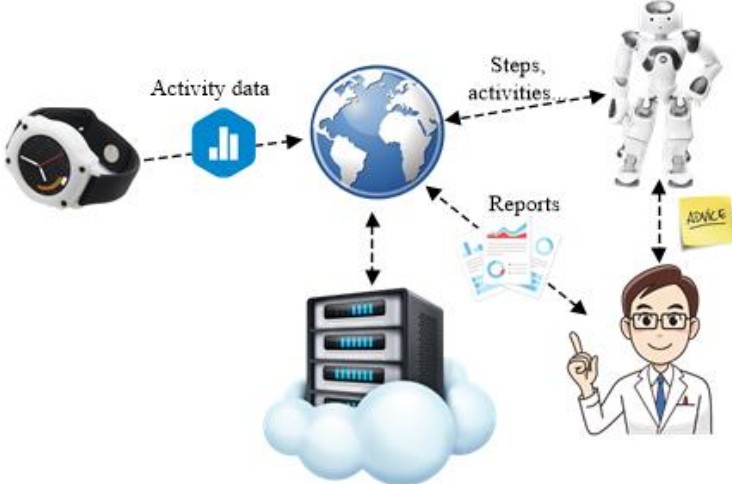

**Figure 5.** The interaction between the obesity management system components.

*3.2. Hardware Design*

In terms of hardware architecture, the developed obesity management system consists of four main units, which include the following:

1. NAO robot: This is an autonomous and programmable humanoid robot developed by Aldebaran Robotics. The NAO robot is an innovative robot deployed in various social applications for treating children with autism, diabetes, and education [23–26]. In this study, the NAO robot is the main component of the framework, and it was employed to interact with the obese child and offer general advice and productive stories.

2. Activity monitoring unit: This includes a low-cost wearable watch (TTGO T-Watch) presented in Figure 6, that can be attached to the obese child's wrist in order to monitor the child activity during the daytime. TTGO T-Watch is equipped with 3-axis acceleration sensor (BMA400) that can be used to monitor the child's daily activities. We developed a piece of code using an Arduino Integrated Development Environment (IDE) to read the data from the TTGO onboard sensors and process and then send the collected data to the Firebase server through a WiFi communication. The developed code was able to measure the everyday activities for obese children, including the steps, active/inactive time, and sleep time.

- Active time refers to the total time the child is active in the daytime, where the child may be walking, playing, running, etc.
- Number of steps refers to the total number of steps that are accomplished by the child during the day.
- Inactive time refers to total time the child is inactive, for instance, sitting down, laying, etc.
- Sleep time refers to the total sleep time for the child during the day.

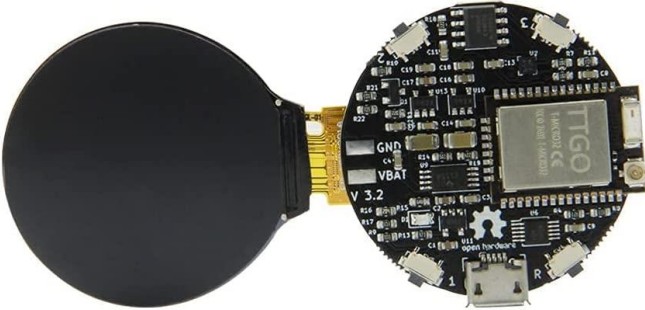

**Figure 6.** TTGO T-watch with ESP32 module.

3. Web-server side: This involves gathering the information received from the smart watch and the robot unit; processing information; and making it available for the medical staff and parents.

4. Medical staff side: This incorporates the individuals who give medical care exhortation and therapy to the obese kid based on the information received from various sensors and the historical information available on the server.

*3.3. Software Architecture*

The main contribution of this paper is based on developing a robot interaction system. In order to achieve this task, the NAOqi programming framework was used to develop the robot interaction system for NAO. The NAO framework is the programming environment that was used to program NAO, Pepper, and Romeo robots. In addition, the NAO framework establishes a homogenous communication between different robot modules, including audio, motion, and video, homogenous programming, and homogenous information sharing. Figure 7 shows the main modules that were employed in the robot system. In the proposed system, the NAOqi architecture consists of the following packages:

1. NAOqi Core: This includes a list of core modules where every module offers a set of methods. The developed robotic system employed a set of core modules that handle

core operations, such as start and stop behaviors and managing connections between different modules.

2.  NAOqi Motion: This package involves several methods that allow the NAO robot to move and perform several actions. For instance, NAO moves hands when it interacts with the obese children. In addition, NAOqi Motion offers a set of functionalities that allow the NAO robot to perform navigation and movement.

3.  NAOqi Audio: This involves the audio software components of the NAO robot platform. In the proposed system, the NAO robot is able to interact with the obese child through a voice recognition system, where the NAO robot may communicate with the obese child using the Arabic language. This also includes the text-to-speech functions and speech recognition functions.

4.  NAOqi Vision: This contains a set of vision components for the NAO robot platform. The developed robot system is able to recognize the identity of a child using a face recognition application. This helps the robot to correctly retrieve the child's activities during the last few days in order to perform the suitable action accordingly.

5.  NAOqi Sensors: This involves a set of modules that allows the developers to interact with the sensors available in the NAO robot. The NAO robot platform is equipped with a set of sensors, including the range-finder sensors that allow the robot to detect the presence of heading objects and tactile sensors that detect whenever the NAO robot is touched by the obese child.

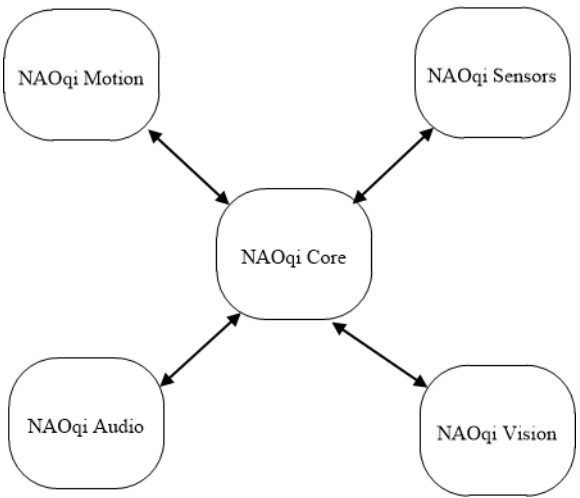

**Figure 7.** Employed NAOqi modules.

NAOqi Motion, NAOqi Sensors, NAOqi Audio, and NAOqi Vision packages are all controlled by the NAOqi Core package. Hence, NAOqi Core is able to start and stop any function in any package in order to perform the suitable action accordingly.

## 4. Results

This section discusses the experiment setup and design and presents the results obtained from several experiments conducted with obese children.

### 4.1. Participants Information

A total of 16 children and their parents were invited to participate in the pilot study conducted in the Industrial Innovation and Robotics Centre (IIRC) at the University of Tabuk. The pilot study consisted of 16 children (11 boys and 5 girls) aged between 6 and 10 years old, and it was conducted over a four-week period. Table 1 shows the participant information.

**Table 1.** General information about the participants (obese children).

| Children # | Gender | Age | Weight | Length | Overweight % |
|------------|--------|-----|--------|--------|--------------|
| 1 | M | 7 | 41 | 119 | 28.9 |
| 2 | F | 8 | 43 | 122 | 28.8 |
| 3 | M | 7 | 45 | 118 | 32.3 |
| 4 | M | 10 | 53 | 144 | 25.6 |
| 5 | M | 9 | 52 | 142 | 26.2 |
| 6 | F | 7 | 38 | 114 | 29.2 |
| 7 | M | 8 | 38 | 120 | 26.4 |
| 8 | F | 10 | 52 | 148 | 23.7 |
| 9 | M | 7 | 36 | 116 | 26.8 |
| 10 | M | 8 | 45.5 | 123 | 30.1 |
| 11 | M | 8 | 43 | 121 | 29.4 |
| 12 | M | 9 | 43 | 126 | 27.1 |
| 13 | F | 10 | 52 | 146 | 24.4 |
| 14 | M | 7 | 41.5 | 116 | 30.8 |
| 15 | F | 8 | 42.5 | 123 | 28.1 |
| 16 | M | 10 | 56.6 | 152 | 24.5 |

According to several studies [27,28], children need at least 60 min of physical activity each day, for instance, running, climbing, swinging on monkey bars, push-ups, sit-ups, lifting weights or yoga, etc., in order to maintain good health.

The obese children and their parents gave informed consent and answered a questionnaire relating to the demographics and medical background of the obese child. The study was conducted for four weeks, during which each child was asked to attend a robot session every two days (three times a week). For instance, child 1 attended a session on Saturday, Monday, and Wednesday. The total interaction time for each child was between 25 and 40 min in each daily session.

*4.2. Experiment Design*

For this study, a personal robot system that interacts with obese children was developed. As soon as the obese child attended the robot session, the robot recognized the child based on a pretrained face recognition model. Then, the robot collected the child's activity for the last 2–3 days from the database and started behaving with the child according to his/her activity. The robot first presented several recommendations for the child according to the obtained activity data from the firebase. Then, the robot performed a quiz session, during which the robot asked the child several questions, and then it stored the score obtained from the sessions in the firebase server.

Every child had a smart-watch attached to his/her arm during the day. However, the parents needed to recharge the smart-watch during the night-time (when the child went to bed). Every child along with his/her parent were invited to visit the Industrial Innovation and Robotics Centre (IIRC) in the University of Tabuk every two days in order to meet the medical staff and to interact with the robot system. At the end of the experiment, parents were asked to fill in a questionnaire along with their obese children.

In general, the robot interacted with each child individually through several stages in each session. The first session (introduction) took approximately 7 min, where the robot recognized the child's identity and welcomed him/her; then, the robot retrieved the child's activity information from the firebase database. The second session involved general advice about the child's activity during the last 2–3 days, and this took approximately

12 min. The third session included completing a general test about the obesity, and this took about 15 min. Finally, the last session involved several physical activities such as performing exercises.

*4.3. System Evaluation*

According to several recent studies [29–35], knowledge plays a significant role in children self-management since the enhancement of children's knowledge may contribute to more effective management and better adherence, and it may enhance the education level for several diseases, such as autism, asthma, diabetes, and obesity. In a study focused on gaming interventions [29], which also attract young people attention engaging in self-management activities, it was identified that knowledge about the illness improved self-management behaviors. Similarly, another study [30] used personal robots for engaging children with quizzes for improving their diabetes self-management knowledge. The study observed this by applying constructive feedback, acknowledging feelings and moods, and encouraging competition. The robotic intervention can help in improving health literacy in children in a pleasurable, engaging, and motivating way and can contribute to the self-management of chronic diseases such as diabetes.

Children' knowledge can be enhanced through providing general information about obesity, exercise, and diet using attractive methods such as video games, mobile applications, and robots. For this study, a robot system was developed and employed as an assistant to obese children.

In the initial stage, the parents were asked about their children's medical background. After each session, the parents along with their children were requested to fill out a questionnaire that consisted of three main parts, weight, food regulation, exercise, and emergency precautions. The answers were filled using a 5-point Likert scale: 1 (I never do) until 5 (I always do).

There were 15 questions about how the obese children felt towards the robot that could be answered on a 5-point Likert scale from 1 (Never) to 5 (Always). These questions were answered after the first four sessions, when the children got used to the NAO robot.

Similar to a recent study [19], the acceptance of the medical social robot was measured using enjoyment, social presence, and social anxiety factors. Accordingly, in this paper, the developed obesity management system was evaluated by assessing the following attributes:

1. Interaction time: This estimates the average interaction time between the obese children and the robot system. Total interaction time should offer positive feedback on the acceptability level of social robots (NAO robot platform in our case).
2. Education quiz: This measures the average results obtained from the education quiz offered by the robot and the knowledge received by the obese children.
3. Enjoyment: This indicates how often the obese child enjoyed interacting with the robot system. Hence, the enjoyment should offer a good indication of the obese children's acceptability level of social robots.
4. Acceptability: This measures the average acceptability of the social robot platform by the obese children during the engagement sessions for the whole experiment time.

Figure 8 shows the average interaction time with the robot system for each obese child. It is evident that most of the children interacted with the robot system for almost the full target period with an average percentage of (88.75%). Therefore, most of the obese children were interested in interacting with the robot system. In addition, the obtained results showed a high acceptability level, as analyzed from the positive comments made by parents about their children's enjoyment and interaction with the NAO robot. However, achieving good interaction levels and acceptability in the long-term can be challenging using robotic systems, as the children may get bored after using it for a certain amount of time. To address such issues, a recent study [36] integrated mobile health applications and used hybrid artificial brain (with technologies such as artificial intelligence (AI) and machine learning (ML)) with a cloud-based robotic system to enable the prolonged use of the system.

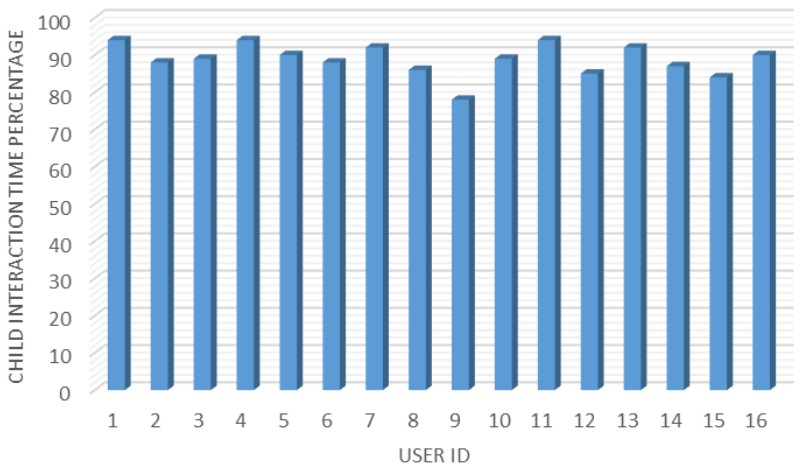

**Figure 8.** Percentage of child interaction time with the robot system.

Enabling approaches such as positive behavioral support using AI and ML techniques can enable robotic systems to adapt its behavior to the child's (and parent's) needs and desires and, therefore, progress with the child [37]. However, a recent systematic review [38] on the efficacy of emerging technologies to manage childhood obesity identified the lack of research on integrating robots and AI to manage obesity in children, although the approach could be effective for treating childhood obesity in the future.

On the other hand, Figure 9 presents the evaluation of the enjoyment activity between the obese child and the robot system. As noticed in the figure below, a high percentage of obese children enjoyed interacting with the robot system, and this may help in maintaining the diets and exercise for obesity children. The average enjoyment percentage for all the 16 children was around 88.18%. Social robots are identified to be generating interest or excitement regardless of what they appear to be like [39], and they are particularly attractive to children [40], as they introduce an element of fun, curiosity, and excitement among children [41]. Assessing the children's enjoyability percentages in relation to the NAO robot (88.18%), it can be analyzed that the NAO robot was attractive and could have been successful in generating fun and excitement among the children. However, as discussed earlier, there is a need to withhold or improve the robotic interaction with children for continued engagement and improved awareness, which achieve self-management goals for obesity.

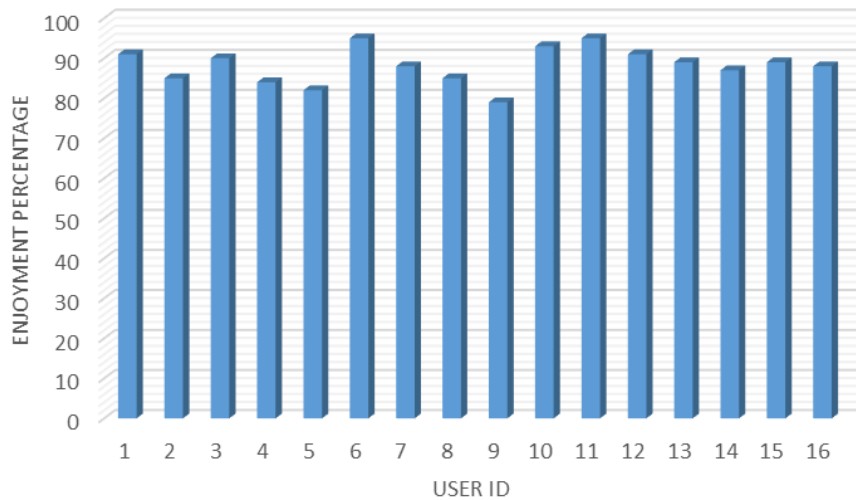

**Figure 9.** Average enjoyment time when the obese child interacted with the robot system.

The robot acceptability issue was also addressed in this work. It is important to study the robot acceptability for children with obesity. Figure 10 shows the average acceptability robot for children with obesity with an average of 89.37%. This study showed that humanoid robots were well accepted by obese children and parents. There was no difference in the overall acceptability of the robot between the male and female obese children (88.68% and 90.07%, respectively). There were no major differences observed in the acceptable levels of male and female obese children, indicating no impact of gender on the acceptability factor, thereby supporting a unified design and approach for the robotic design for obese children self-management practices.

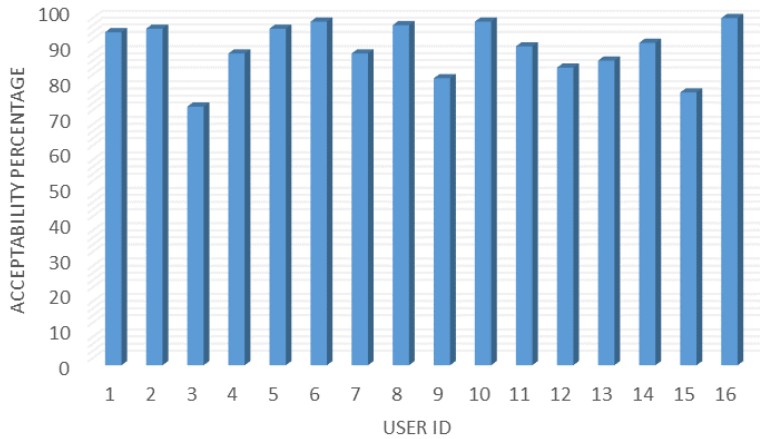

**Figure 10.** Robot acceptability percentage for the 16 users.

Regarding the quiz service, obese children answered 12 out of 16 questions on average. The obese children thought the robot system and the quiz expressed fun and motivating issues. Although the children reflected good scores in the quiz conducted by the NAO robot, there were a few aspects that might have led to lower scores among user 3 and user 15 when compared to other users. It is important to identify these factors that may influence the child learning process. It is important for the robot to assess the child's knowledge level, and accordingly, it should select the topics that are not too difficult nor too easy but are at the right level of difficulty customized for each child [36,42]. Figure 11 presents the education quiz percentage for the 16 users. Such an approach allows the child to be in his/her "zone of proximal development," which is known to provide an optimal educational path, allowing the knowledge acquirement process to be according to the children's skills and abilities [43].

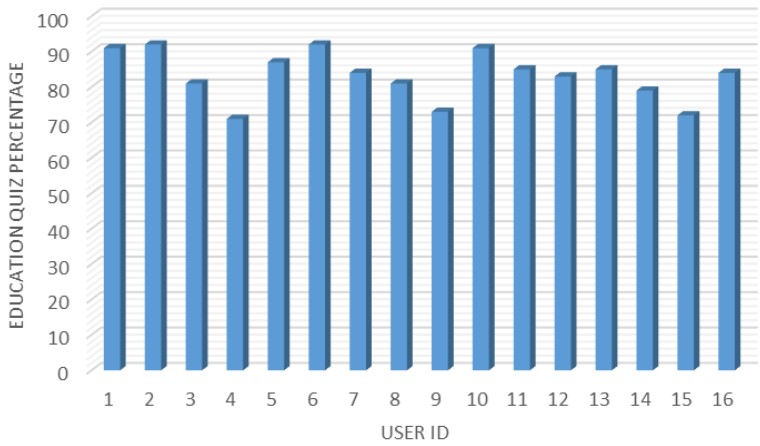

**Figure 11.** Education quiz percentage for the 16 users.

As presented in the recent study [38], twenty-six technological interventions for childhood obesity prevention and treatment were discussed and analyzed, where authors revealed that there was a lack of research studies using the AI to manage obesity in children. Mainly, various technologies have been used for the purpose of managing childhood obesity, including mobile-based interventions, game-based interventions, and web-based interventions. Only five research studies employed AI technology in their obesity investigation with children. However, none of these studies employed social robots for the management of children with obesity.

The obese children's average acceptability level of the recently developed childhood obesity management systems [20–22] was around 82–87%. However, the proposed obese management system in this paper offered a reliable acceptability level of 89.37%. Therefore, a robot is a promising technology that can further enhance obesity management for children with obesity.

Similarly, an IoT-enabled robotic assistant for diabetes management among the children achieved 85–90% acceptable levels, which was found to be effective in promoting self-management practices [44]. Thus, the NAO robot can also be effective in promoting obesity self-management practices by improving the children's knowledge and awareness. However, there were some issues identified with the NAO robot in a similar study for diabetes self-management among children [45]. The study [42] found that a few movements of the robot were entertaining, but not all children appreciated the eye contact and expressions contributed to bonding. Therefore, there is a need to adopt more methods such as interviews and focus groups to extract more quality information from the children in relation to the NAO robots for assessing the acceptability levels in detail.

The obtained results are promising, and a number of improvements were identified in order to incorporate them in the planned field study. As a result, this paper shows a study that provides a strong indication for how robot systems can enhance the children's perceived enjoyment of learning and help them to learn more about obesity.

## 5. Conclusions

In this study, we focused on investigating the acceptability of a social robot with obese children. According to our knowledge, this is the first study conducted in this field that employed education through a child-robot interaction and that monitored using a smart IoT device. Moreover, this study showed how a personal robot can assist obese children in managing their exercises and diets in a relatively enjoyable way. A pilot clinical study was conducted with the goal of exploring how obese children and their parents assessed the proposed platform. The obtained results showed a high acceptance rate for the developed robot system, along with the education system. In addition, the obtained results in terms of the acceptability level of the proposed system outperformed the results achieved in recent similar studies. The obese children complied with the presented study procedure, and they were positive about the intervention. The employed NAO robot along with the smart watch device were suitable for measuring fun, motivation, and acceptability. Various potential improvements in robot behaviors and the evaluation method were identified.

**Author Contributions:** M.A. studied and analyzed the recently developed systems. In addition, M.A. conducted the pilot study on 16 different children and discussed and analyzed the results. On the other hand, T.A. developed the robot system in terms of interaction functionalities. Moreover, T.A. implemented the smart-watch code, the robot code, and the webserver framework. All authors have read and agreed to the published version of the manuscript.

**Funding:** This research project was funded by the Deanship of Scientific Research (DSR) at the University of Tabuk, Tabuk, Saudi Arabia, under grant no. [1440-258].

**Acknowledgments:** The authors would like to acknowledge the financial support for this work from the Deanship of Scientific Research (DSR) university of Tabuk, Tabuk, Saudi Arabia, under grant no. [1440-258].

**Conflicts of Interest:** The authors declare no conflict of interest.

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
