# Peer review of "Design and Evaluation of a Personal Robot Playing a Self-Management for Children with Obesity"

_electronics, doi:10.3390/electronics11234000_

Round 1

Reviewer 1 Report

This work addresses the acceptability of social robot with obese children. The theme is interesting, novel, and the article is generally well written.

Some improvements:

- The smart-watch hardware unit that incorporates this system should be described with more detail.

- The quality of some figures is low (ex: figure 2 and figures 8 to 11).

- The NAOqi programming framework must be better described, as well as all the robot interactions.

- Authors should give more focus on how the experimental results were computed, in order support the conclusions.

Reviewer 2 Report

The authors addressed an important health issue at a young age. They presented a comprehensive background and good experimentation. I want to know if there is any other work like this in the past that can be considered as state of the art so that the proposed work can be compared and gauged with the state of the art. This way the proposed work can be established more comprehensively.

1. What is the main question addressed by the research?
This research addressed a very important health issue, obesity, particularly in children. The authors proposed a unique way with the help of an NAO robot to engage children in activities and help them to overcome obesity.

2. Do you consider the topic original or relevant in the field? Does it address a specific gap in the field?
The topic does address this problem in a different way, but of course, there are some existing systems already which addressed this problem but in somewhat different ways.

3. What does it add to the subject area compared with other published material?
In this research, authors have developed a social robot system that engages children to activities while interacting with a robot to help them with reducing obesity. They also validated this model using experimentation.

4. What specific improvements should the authors consider regarding the methodology? What further controls should be considered?
As i mentioned earlier, in order to establish this research work robustly, authors must compare their work with other state-of-the-art works. This will help more objectively to gauge the performance of the proposed model.

5. Are the conclusions consistent with the evidence and arguments presented and do they address the main question posed?
The conclusion is ok, but once they add the comparison above, they should update the conclusion part accordingly.

6. Are the references appropriate?
References are fine in this form, but they have to update them as well once they include some works for comparison.
